# Does Digital Inclusive Finance Mitigate the Negative Effect of Climate Variation on Rural Residents’ Income Growth in China?

**DOI:** 10.3390/ijerph19148280

**Published:** 2022-07-06

**Authors:** Chunyan He, Anjie Li, Ding Li, Junlin Yu

**Affiliations:** 1School of Economics, Xihua University, Chengdu 610039, China; 1220200034@mail.xhu.edu.cn (C.H.); l1160708207@sina.com (A.L.); 2School of Public Administration, Southwestern University of Finance and Economics, Chengdu 611130, China; 3School of Business Administration, Yonsei University, Seoul 03722, Korea; yujunlin98@yonsei.ac.kr

**Keywords:** climate variation, digital inclusive finance, rural residents’ income growth, poverty alleviation, rural revitalization

## Abstract

Global anthropogenic greenhouse gas emissions have exacerbated climate variation. Climate variation impacts the agricultural production and rural residents’ income negatively, further widening the urban-rural income gap and harming the co-benefits. Narrowing the income gap has always been a global concern and an important part of China’s rural revitalization strategy. However, little is known about whether digital inclusive finance can mitigate the negative impact of climate variation on rural residents’ income growth in China. Using panel data from 31 provinces in China from 2011 to 2019 and a digital inclusive finance index developed by Peking University, together with historical temperature data, this study examined the impact of digital inclusive finance on Chinese rural residents’ income growth in response to climate variation. It was found that digital inclusive finance could promote rural resident operating, wage, and transfer income growth. A heterogeneity analysis revealed that rural residents in central and western regions experienced larger digital inclusive finance facilitating effects on income growth than the eastern regions. Further analyses using the Spatial Dubin Model found that digital inclusive finance had a spatial spillover effect as it could significantly promote income growth in neighboring provinces. Although climate variation reduced rural residents’ income and increased their risks, digital inclusive finance significantly mitigated this negative effect. Digital information infrastructure construction, financial risk prevention, digital financial knowledge, and e-commerce popularization were practical paths to optimizing inclusive finance development in rural areas and promoting poverty alleviation and rural revitalization to resist climate risks.

## 1. Introduction

Climate variation impacts the global economy deeply and has become one of the greatest threats to sustainable human development. Greenhouse gases emitted by human activities cause global warming and aggravate climate variation, and rising temperatures will increase sea levels [1], resulting in more severe and frequent natural disasters [2]. Moreover, aggravation of climate variation will make global extreme climate events more frequent. Extreme weather such as high temperature, heat waves, droughts, floods, hurricanes, cold waves, and other extreme weather have extensive impacts on human health, agriculture, economy, and natural ecosystems, and these impacts will lead to significant welfare losses for individual families [3,4]. According to the “2019 Global Climate Risk Index Report”, since 1998, 526,000 people have died from extreme weather in the world, and the direct economic loss has reached USD 3.47 trillion. The more vulnerable individuals, groups, classes or regions are, the more easily they are affected by the environmental shocks brought by climate variation [5]. Vulnerable groups such as rural women, children, and herdsmen have limited access to land, employment, and public services, and their ability to cope with climate variation risks is also weaker. For example, women who experienced drought as children are shorter, have poorer economic status, and are less educated [6,7]. Climate variation has a huge impact on rainfall-dependent agricultural industries and may lead to unstable incomes for rural residents [8]. Poverty is related to income levels and vulnerability. Poor farmers are particularly vulnerable to income risks, which can limit agricultural production and farmers’ agricultural investment and lead to poverty traps [9].

Rapid digital finance development can provide new opportunities for rural resident income growth as it can improve capital circulation, promote the equalization of opportunities, and reduce capital costs [10]. However, current rural digital inclusive finance faces several difficulties, such as a lack of digital financial knowledge and the insufficient use of rural digital financial products [11]. Further, there have been few studies that have examined the direct and indirect effects of digital inclusive finance on rural resident income or assessed the benefits rural residents can gain from digital finance as part of China’s rural revitalization strategy. 

As the income gap between urban and rural residents affects resident happiness and can negatively impact social-economic development and socio-political stability, governments around the world have been working hard to narrow this gap [12]. Urban-rural income inequality indicates an imbalance in China’s social-economic development [13]. Besides the significant income gap between urban and rural residents in China (Figure 1), rural residents in different regions also have obvious income gaps (Figure 2). 

As digital technology and finance become more deeply integrated into everyday society, digital finance is reshaping financial markets and the competitive landscape and becoming more inclusive in both urban and rural sectors. Compared with digital finance, traditional finance is unsustainable and does not support rural resident income growth [14]. As China’s rural residents are under significant pressure to increase their incomes, they urgently need to find new income opportunities [15]. Digital inclusive finance brings them the new opportunities, especially for small and medium-sized enterprises [16] and low-income groups such as rural residents [17].

China’s social and economic activities have new networking and digitization characteristics [18]. The “Analysis of the Digital Economy Situation in 2020–2021” issued by the China National Industrial Information Security Development Research Center reported that China’s digital economy accounts for more than one-third of China’s GDP. Digital finance has helped promote the rapid expansion of the new digital economy, accelerated the transformation of the traditional financial system, and augmented inclusive financial processes. The “46th China Internet Network Development Statistics Report” reported that by June 2020, there had been 850 million online payment users, 196 million to 244 million of whom were rural online payment users, or 35.97–44.82% of China’s rural population. Various agricultural-related online lending platforms are providing hundreds of billions of yuan in credit funds to rural areas in many provinces, large enterprises have established Internet-based financial platforms, and the traditional agricultural financial institutions, such as the Agricultural Banks and Rural Commercial Banks, are also accelerating their online financial businesses. Digital inclusive finance development relies on digital infrastructure and consumer awareness. Compared with urban areas, there are still some limitations in rural digital inclusive finance. In some rural areas of developing countries, as the Internet infrastructure is poor and power supplies unstable, the rural digital finance and e-commerce development is lagging behind. Moreover, most rural residents lack basic financial and Internet knowledge, which restricts their participation in digital inclusive finance. Some empirical studies have found that the financial needs of rural residents with less access to the Internet infrastructure have not been effectively stimulated, and even their lives have become more embarrassing [19,20]. Therefore, as digital finance is showing strong vitality, studying how the digital inclusive finance impacts the rural residents’ income and whether digital inclusive finance mitigate the negative effect of climate variation on rural residents’ welfare has a certain reference value for countries in which there are large rural–urban income gaps.

Digital inclusive finance continues to develop rapidly in China and its impact on rural residents’ welfare has been widely concerned. Little attention has been paid to promoting rural residents’ income growth under climate variation. It is significantly practical to study whether digital inclusive finance can promote rural residents’ income growth in China, thereby reducing the negative impact of climate variation. This study can help reduce poverty caused by climate variation and promote rural revitalization in developing countries. Using panel data from 31 provinces in China from 2011 to 2019 and a digital inclusive finance index developed by Peking University, together with historical temperature data, we empirically analyze the impact of digital inclusive finance on income growth by China’s rural residents for a new climate variation mitigation strategy. Other contributions of this study are as follows. (1) Compared with the case studies on the impact of digital inclusive finance on the income of residents in the existing literature, this study pays special attention to its impact on rural residents’ income, which could provide new strategies for rural poverty alleviation and urban–rural income gap narrowing in developing countries. (2) Climate variation will further aggravate the poverty and vulnerability of vulnerable groups, reduce welfare, and affect regional economic growth. This study combines socioeconomic data and natural data to test whether digital inclusive finance can mitigate the negative impact of climate variation on rural residents’ incomes growth, and increase the co-benefit which refers to the non-climate benefits arising from greenhouse gas mitigation strategies, which has not been covered in the existing literature.

## 2. Literature Review

### 2.1. Climate Variation and Rural Residents’ Income Growth

The ecological environment impacts the economic growth and regional sustainable development deeply [21,22]. Human-induced greenhouse gas emissions led to climate variation, and extreme climate events are more frequent around the world. From a global perspective, high temperatures and extreme weather can slow the global economy by 0.25% [23]. Ciscar et al. (2011) found that climate variation could halve the annual wealth growth rate of European residents by 2080 [4]. Continued climate variation has a huge impact on rainfall-dependent agriculture [24], thereby slowing down the growth of rural resident’s income.

Some scholars have studied the relationship between the climate variation and residents’ income. Chen et al. (2020) found that the climate variation would reduce the farmers’ disposable income and increase the urban–rural income gap in China [25]. In terms of agricultural production input, according to climate variation forecasts, farmers will change crop varieties and increase investment in climate disaster prevention, which will lead to an increase in agricultural production costs and total inputs and decrease in farmers’ net income growth [26]. In terms of output, changes in light conditions, heat sources, water resources, etc., which are manifestations of climate variation, affect soil conditions. Crop yield and quality are associated with soil conditions [27]. Poor soil conditions will inevitably reduce agricultural output and farmers’ income [3].

### 2.2. Digital Inclusive Finance and Rural Resident Income Growth

To break the vicious rural poverty circle, it is necessary to strengthen financial product availability in rural areas, continuously expand investment, and promote rural capital accumulation [28]. Digital inclusive finance, which is the combination of digital technology and financial services, has stronger temporal and spatial penetration [29]. Digital inclusive finance improves the circulation of the rural financial system and the new technologies adoption of rural residents, and promotes innovation of management and new technologies and digital upgrade of rural industries, which indirectly increases rural residents’ income and welfare [30]. For example, digital inclusive finance can help the commercial banks establish a more comprehensive data sharing system and a big data credit risk prevention system. Thus, commercial banks can obtain more market information and provide better credit services for rural residents [31]. Then, better credit services can help rural residents scientific manage their assets, improve their risk resilience, and indirectly increase their income levels. Using cross-sectional data from 93 developing countries in 2017, Asongu and Nwachukwu (2017) examined the impact of mobile banking on developing country poverty alleviation and income inequality, finding that mobile banking engendered inclusive financial development [32], and Uduji et al. (2019) found that electronic wallet technology assisted Nigeria’s smallholder farmers to gain greater financial support [33].

Digital inclusive finance can improve rural residents’ risk response capabilities [34], and also has significant medical, income, and education poverty alleviation effects, helps family farms increase their use of green technologies by improving their credit availability [35], promotes rural household consumption, especially in households with fewer assets and lower incomes, and effectively bridges the income gap between urban and rural residents [36]. In the central and western regions of China, in which there is low urbanization and low physical capital, the effects of digital finance on economic growth could be more significant [37]. 

While previous studies have examined the relationships between digital finance and rural income growth, there have been few studies that have rigorously analyzed the impact of digital inclusive finance on rural resident income growth and the spatial spillover effects. Therefore, it has been difficult to clarify the value of the digital inclusive finance mechanism on rural resident income growth and income structure optimization or to propose effective and feasible policy options that fully consider the various constraints under temporal and spatial heterogeneity. More importantly, there is no direct evidence between digital inclusive finance and rural residents’ ability to minimize the risks of climate variation. To fill these research gaps, this study adopts panel data from 31 provinces in China from 2011 to 2019 and a digital inclusive finance index developed by Peking University, together with historical temperature data to explore the spatial characteristics of digital inclusive finance and its impact on the income growth of rural residents under climate variation from the perspective of climate change mitigation strategy.

## 3. Theoretical Hypothesis

**Hypothesis** **1.**
*Digital finance is convenient, low cost, and has wide coverage and sustainability, which are conducive to reducing information asymmetry and the financial service threshold constraints of disadvantaged groups, increasing inclusive rural financial services demand and supply levels, and promoting rural resident income growth.*


Internet speeds now allow digital financial services to be handled using mobile devices, as the transaction processes are standardized, the use procedures simplified, and the service methods convenient. Timely financial information can be easily found on the Internet, which reduces the costs of learning about financial services.

The “ubiquitousness” of the Internet means that digital finance has almost zero marginal costs, that is, the financial services threshold is significantly lower because of lower market transaction maturity matching costs. Compared with traditional payments, online payments have no time and space limitations. As digital financial services can be accessed through mobile devices such as smartphones and tablet computers, rural residents in remote areas can access efficient, fast payment services, which increases the financial supply for small enterprises, rural residents, and other low-income groups that have had difficulties in obtaining financial services from traditional financial institutions. However, there is also some limitations in digital finance. A convenient, safe, and instant digital payment network relies on sound digital infrastructure and consumers’ cognitive abilities. Therefore, digital finance development needs systems such as education, technology, and finance, which are synergistic. If their synergy fails, the digital finance may be ineffective, and even emerge new financial exclusion.

**Hypothesis** **2.**
*Rural resident income growth from digital inclusive finance availability has a spatial spillover effect.*


Under the combined effects of transportation costs, language, customs, diet, and other geographic relationship factors, the imitation learning costs between similar areas are relatively low as similar social-economic conditions make imitation learning more likely to succeed. Population migration, and especially cross-regional rural population migration, has strengthened the correlations between rural resident income growth in similar regions. Modern information technological development has lowered information barriers, promoted cross-regional social and economic links, and strengthened the inter-relevance of regional rural resident income growth.

Digital finance and digital economic development, therefore, are conducive to knowledge spillovers as they can reduce the labor communication and coordination costs of geographical distance, making it easier to transfer local economic activities to other regions. Digital Internet-based finance technologies provide a networked economy that links the financial and economic activities of various regions. Through diffusion effects, areas with high economic development can help drive economies with low development. Development is also accompanied by rural non-agriculturalization and urbanization, which have positive spatial spillover effects on rural resident income growth. However, digital finance accessibility may also result in negative spatial spillover effects because digital finance reduces the barriers to cross-regional capital, information, and technology, which means that various production factors can quickly and easily flow to high-yield areas, generating agglomeration effects. Therefore, the development of one region may come at the cost of a decline in another region.

**Hypothesis** **3.**
*Climate variation has a negative impact on rural residents’ income growth, but digital inclusive finance can mitigate this negative impact.*


Climate variation has led to frequent extreme weather events that have a direct impact on rainfall-dependent agriculture. For agricultural production inputs, climate variation increases agricultural production costs and total inputs. For outputs, climate variation has altered soil conditions, further affecting farmers’ income. Therefore, climate variation increases the instability of agricultural production, reduces the rural residents’ income, widens the urban-rural income gap, and creates the poverty trap. Digital inclusive finance can promote the development of rural finance to improve the efficiency of resource allocation. The effective resource allocation can reduce the agricultural production costs caused by climate variation. Digital finance provides rural commercial banks with a clearer and more comprehensive data sharing system, better credit services, and more accurate and detailed planning and arrangements, which gives more financial support to rural residents and raises the agricultural productivity, thereby reducing the instability of income caused by climate variation.

## 4. Data and Methodology

### 4.1. China’s Digital Financial Indicator System

Compared with its digital finance development, China’s digital financial indicator system construction is lagging. The currently available digital financial indexes are the “Peking University Digital Inclusive Finance Index of China (PKU-DFIIC)”, the “Peking University Internet Financial Sentiment Index”, and the “Commercial Bank Internet Transformation Index” issued by Peking University. The digital financial indicator used in the empirical part of this article was the “Peking University Digital Inclusive Finance Index of China”.

The PKU-DFIIC is based on massive transaction data from the Ant Financial accounts that is compiled using an analytic hierarchy process coefficient variation weighting method. The PKU-DFIIC can be compared in both regional and time dimensions, reflects multiple digital financial service levels, and has diversified features; therefore, as it has coverage breadth, usage depth, and a good digitization level. The PKU-DFIIC has good usage depth because it has various sub-indexes such as payment, credit, insurance, investment, and funds, has good time continuity, and three digital finance levels; provincial, city, and county. As the data are of high quality, it is possible to comprehensively study digital financial service development in China and its associated economic effects, which is why the PKU-DFIIC is one of the few digital financial indexes that has been widely adopted by researchers.

All other data were taken from the China Statistical Yearbooks and the China Agricultural Statistical Yearbooks.

### 4.2. Variables and Benchmark Regression Model

Digital inclusive finance has many advantages. It is low-cost and convenient and has wide coverage and sustainability, which reduces information asymmetry, lowers the financial service threshold constraints faced by disadvantaged groups, improves the demand and supply of rural financial services, and increases financial service inclusiveness, all of which can promote rural resident income growth.

2011–2019. provincial panel data were used to test the impact of digital inclusive finance on rural resident income growth and income structure and to analyze regional difference heterogeneity. The benchmark regression equation was as follows:(1)$$\begin{array}{c} ln{\left(IN\right)}_{it}=\beta_{0}+\beta_{1}ln{\left(DFIIC\right)}_{it}+\beta_{2}AMA_{it}+\beta_{3}FU_{it}+\beta_{4}GSA_{it}+\beta_{5}FSA_{it}+\beta_{6}TRADE_{it}+\\ \beta_{7}ln{\left(perGDP\right)}_{it}+\beta_{8}UR_{it}+\beta_{9}TI_{it}+\mu_{i}+\varepsilon_{it} \end{array}$$
where *i* represents the province and *t* represents the year. The explained variable IN denotes the per capita rural resident disposable income, and DFIIC is the digital inclusive finance index explanatory variable used to measure the rural digital inclusive finance development level. Previous studies found that agricultural machinery usage, fertilizer usage, and financial support for agriculture and trade openness impacted rural resident income growth [38]. Therefore, based on previous studies, this study chose the following control variables: The ratio of total agricultural machinery power to crop planting area; the ratio of agricultural fertilizer usage to crop planting area; the ratio of fiscal expenditure on agriculture, forestry, and water affairs to agricultural GDP; the ratio of agricultural loans to agricultural GDP; the ratio of total imports and exports to GDP; the GDP per capita; the urbanization rate; and the tertiary industry proportion. $\beta_{0}$ is a constant term, $\mu_{i}$ is the fixed effect, and $\varepsilon_{it}$ is an error term. Table 1 and Table 2 show the variable definitions and descriptive statistics.

### 4.3. Spatial Autocorrelation and the Spatial Dubin Model (SDM)

The existence of a spatial correlation between samples is a prerequisite for spatial econometric analysis. If there is no spatial correlation between the samples, there is no need to construct a spatial model as a general model is suitable. Moran’s I is the most popular statistical test for spatial autocorrelation [39], with global spatial autocorrelation usually employing Global Moran’s I, which examines whether there is spatial correlation over an entire region.

The hypotheses state that the impact of digital finance on rural resident income growth has spatial spillover effects. The above spatial correlation analysis found that there was a significant positive spatial autocorrelation between digital finance and rural resident income growth. Therefore, a spatial panel model was constructed to test this spatial spillover effect. Elhorst (2014) claimed that almost all panel data models could be directly estimated using the Spatial Dubin Model (SDM) [40]. Compared with spatial autoregression (SAR) and spatial error models (SEM), SDM has significant advantages. First, SDM is the general form of the SAR and SEM, has fewer constraints, and has stronger applicability. Second, the SDM considers both the spatial autocorrelation of the dependent variable and the residual term and the influence of the interactions between the independent variables and the dependent variable, that is, it provides more convincing results. The SDM is as follows:
(2)$$\begin{array}{cc} \; & \ln{\left(IN\right)}_{it}=\alpha_{0}+\alpha_{1}ln{\left(DFIIC\right)}_{it}+\alpha_{2}AMA_{it}+\alpha_{3}FU_{it}+\alpha_{4}GSA_{it}+\alpha_{5}FSA_{it}+\\ \; & \alpha_{6}TRADE_{it}+\alpha_{7}ln{\left(perGDP\right)}_{it}+\alpha_{8}UR_{it}+\alpha_{9}TI_{it}+\beta_{1}ln{\left(DFIIC\right)}_{it}*W+\beta_{2}AMA_{it}*\\ \; & W+\beta_{3}FU_{it}*W+\beta_{4}GSA_{it}*W+\beta_{5}FSA_{it}*W+\beta_{6}TRADE_{it}*W+\beta_{7}ln{\left(perGDP\right)}_{it}*\\ \; & W+\beta_{8}UR_{it}*W+\beta_{9}TI_{it}*W+\rho W*\ln{\left(IN\right)}_{it}+\mu_{i}+\varepsilon_{it} \end{array}$$
where W is spatial adjacency matrix, $\alpha_{0}$ is a constant term, $\mu_{i}$ is the fixed effect, and $\varepsilon_{it}$ is an error term. In SDM, the influence of geographic location is represented by the spatial weight matrix. The spatial adjacency matrix is the most commonly used spatial weight matrix, which indicates that the spillover effect of a region will not only affect the neighboring regions, but also the neighbors of neighboring regions. The definitions of other variables are the same as Formula (1).

### 4.4. Climate Variation Mitigation Effect Model

This section tests whether the digital inclusive finance could mitigate the negative effects of climate variation on rural residents’ income growth. Temperature is a good indicator of climate variation among all climate factors [8]. Katz and Brown (1992) argued that climate variability was more important than the mean temperature [41]. Therefore, the standard deviation of the monthly average temperature over 10 years was used to measure climate change Our analysis focused on panel variability, and a coverage period of 10 years provided a sufficient description of a particular region’s climate.

We used data from the National Bureau of Statistics of China from 2002 to 2019. This dataset reports the monthly temperature of 31 major cities in China (including all provincial capital cities). Here the temperature of the capital city was used to represent the temperature of the province. The monthly temperature is recorded as $T_{c,t}^{m}$ (*m* = January, February, …, December; *t* = 2002, 2003, …, 2019; *c* = Beijing, Tianjin, Shanghai, Hebei, Shanxi, Inner Mongolia, Liaoning, Jilin, Heilongjiang, Jiangsu, Zhejiang, Anhui, Fujian, Jiangxi, Shandong, Henan, Hubei, Hunan, Guangdong, Guangxi, Hainan, Chongqing, Sichuan, Guizhou, Yunnan, Tibet, Shaanxi, Gansu, Qinghai, Ningxia, Xinjiang). The annual average temperature for each province is calculated as ${\bar{T}}_{c,t}$ = $\frac{1}{N}\underset{m}{\sum }T_{c,t}^{m}$, where *N* = 12. The temperature variations of each province in each year are calculated by using the data of the previous 10 years. Climate variation, as the long-term temperature variations over 10 years, can be calculated as follows:$$\begin{array}{c} \;{\mathrm{climate}}_{c_{t_{2011}}}=\mathrm{STD}\left({\bar{T}}_{c,2002},{\bar{T}}_{c,2003},\ldots ,{\bar{T}}_{c,2011}\right)\\ \;{\mathrm{climate}}_{c_{t_{2012}}}=\mathrm{STD}\left({\bar{T}}_{c,2003},{\bar{T}}_{c,2004},\ldots ,{\bar{T}}_{c,2012}\right)\\ \;\ldots \\ \;{\mathrm{climate}}_{c_{t_{2019}}}=\mathrm{STD}\left({\bar{T}}_{c,2010},{\bar{T}}_{c,2004},\ldots ,{\bar{T}}_{c,2019}\right) \end{array}$$

The function STD (·) denotes the standard deviation of the variables in brackets.

A high variable value for a province indicates greater climate variation. Figure 3 shows a map of the temperature variations in the 31 provinces in 2019. More comfortable living is expected in areas with less climate variation.

Combining climate variation data and socioeconomic data, the impact of climate variation on rural resident income growth and income structure is tested. Then the development level of digital inclusive finance is divided into three groups: low level of digital inclusive finance, medium level of digital inclusive finance and high level of digital inclusive finance. The effects of climate variation on income growth will be examined in each group to verify the role of digital finance in moderating climate variation.

## 5. Results

### 5.1. Impact of Digital Inclusive Finance on Rural Resident Income Growth

The Hausman test rejected the random-effects model (RE). Therefore, a fixed-effects model was adopted, the results for which are shown in Table 3. Digital inclusive finance (lnDFIIC), coverage breadth (ln CB), usage depth (lnUD), and digitization level (lnDL) were all found to have significant positive correlations with rural resident income growth, with the respective elastic coefficients being 0.137, 0.088, 0.111, and 0.087. As the explanatory variables may have had a lag effect on rural resident income growth and financial variable time lag effects are usually relatively short, the digital financial variables and all control variables were lagged by one period to estimate rural resident income. Models (5)–(8), found that digital inclusive finance and its coverage breadth, usage depth, and digitalization had a significant lagging effect on income growth, with the respective elastic coefficients being 0.097, 0.061, 0.054, and 0.079. Digital finance was found to have a significant role in promoting rural resident income growth in the current period, the lagged period, the total index, and the index with various dimensions.

The estimated control variable results showed that the estimated coefficient for the use of agricultural machinery (AMA) was not significant, indicating that agricultural mechanization had a limited role in promoting rural resident income growth. In recent years, China’s agricultural machinery service level has increased, which has resulted in an increase in the use of agricultural machinery, a greater division of labor, and greater agricultural production specialization; however, only a small number of large households that provide these specialized services have benefitted from agricultural mechanization as most farmers need to pay for these professional services. The estimated coefficients for government and financial agricultural support were significantly positive.

Before 2011, digital inclusive finance in most parts of China did not occur, so the data of Peking University Digital Inclusive Finance Index of China (PKU-DFIIC) is available as early as 2011. Furthermore, all other data were taken from the statistical yearbooks, which are officially released by the Bureau of Statistics. The latest data is for 2020. To ensure the robustness of the results, that is, whether there will be different trends in the impact of digital inclusive finance on rural residents’ income growth after 2019, we also used the panel data from 2011 to 2020 to test, and found that the results are consistent with the period of 2011–2019 (Appendix A). Therefore, 2011–2019 provincial panel data were used in this study.

Digital inclusive finance can provide financial services production and operating support for new agricultural activities in industrial agricultural enterprises, farmer cooperatives, large professional households, and family farms as it allows them to expand production scales and improve business performances, which in turn increases wage and business incomes. Online financial services can ease the financial constraints on rural entrepreneurial activities, increase non-agricultural employment opportunities, support innovation and entrepreneurship, and create jobs. Online insurance and critical illness crowd funding can also enhance the ability of rural residents to resist risk. When families encounter major difficulties, they can easily obtain transfer payments from relatives, friends, and society; therefore, digital finance can have an all-round impact on rural resident income growth.

Table 4 shows the estimated model results for digital inclusive finance and rural resident operating, wage, property, and transfer incomes. As can be seen, the estimated digital inclusive finance coefficients for the operating, wage, and transfer incomes were significantly positive, with the strongest effect being on transfer income followed by wage income and operating income; however, digital inclusive finance had a negative impact on property income. The results in Table 5 show that the estimated coverage breadth, usage depth, and digitization level coefficients on rural resident operating, wage, property, and transfer incomes were all positive and passed the significance test, indicating that in addition to property income, finance had a positive impact on other rural resident income growth.

### 5.2. Robustness Check

To reduce the digital financial development index endogenous problems due to missing variables, mutual causality, and measurement errors, the lagging terms for the digital inclusive finance total index and each of the dimension indexes were used as the instrumental variables to control endogeneity. Then, a two-stage least squares (2SLS) method was used to test the robustness, the results for which are shown in Table 6. The endogeneity test rejected the null hypothesis at a 1% significance level, which indicated that there was endogeneity and the choice of instrumental variables had been appropriate. Even when the endogeneity was considered, the positive effects of digital inclusive finance coverage, usage depth, and digitization on rural resident income growth were still robust, which confirmed that the development of digital finance has become an important driving force for rural resident income growth.

### 5.3. Heterogeneity Results

The digital financial development impact on rural resident income growth varies because of regional social and economic differences. Table 7 shows the digital inclusive finance impacts on income growth were significantly positive at a 1% level in China’s eastern, central, and western regions, with the respective estimated coefficients being 0.111, 0.142, and 0.135. As can be seen, digital inclusive finance availability had the strongest effects in the central and western regions, both of which have lower economic development. These results indicate that the introduction of digital inclusive finance services can go some way to reducing regional rural resident income gaps and can assist in alleviating poverty and revitalizing rural areas. The estimated coefficients for each region when lagged one period were also significantly positive at a 1% level.

### 5.4. Spatial Spillover Effects

The existence of spatial correlations between samples is a prerequisite for spatial econometric analysis, with the most common spatial correlation method being the Moran index (Moran’s I). In spatial measurements, the weight matrix W is constructed to reflect the spatial relationships between the variables; therefore, choosing a reasonable space weight matrix is vital. For this study, a spatial weight matrix based on geographic relationships was constructed. Anselin (1988) proposed three criteria for establishing spatial weight matrices based on geographic relationships; contiguity, distance [42], and k-nearest. China’s unique geography means that spatial weight matrices based on distance or k-nearest methods are unreliable because the k-nearest method destroys the inherent geographical structure of the provincial spatial units and is unable to accurately quantify the provincial unit spatial relationships. Therefore, a contiguity standard was used to construct the binary spatial weight matrix, after which the matrix was standardized to ensure that the sum of the weight matrix row elements was equal to 1. The Moran index test results showed that the index for income, digital finance development, coverage, usage depth, and digitalization was significantly positive at a 1% level, that is, rural resident income and digital inclusive finance had significant spatial agglomeration characteristics, with the shorter the distance between provinces, the stronger the spatial correlations (Figure 4, Figure 5, Figure 6, Figure 7 and Figure 8).

It was also surmised that the impact of digital finance on rural resident income growth could have spatial spillover effects. Therefore, to test this supposition, a Spatial Dubin model (SDM) was conducted to study the spatial spillover effects associated with digital finance development and income growth. The influence of the independent variables on the dependent variables was decomposed into direct effects and indirect effects to better analyze the spatial interactions between the variables. The direct effect measured the impact of digital finance on provincial income growth, and the indirect effect measured the impact on the neighboring provinces. Because spatial adjacency matrices are better able to capture the rural resident income and digital inclusive finance spatial agglomeration, a spatial adjacency matrix was used in these models. The SDM estimation results (Table 8) showed that under the spatial adjacency matrix, the direct effect coefficient for digital inclusive finance on income growth was significantly positive, indicating that every 1% increase in the digital inclusive finance index led to a 0.327% increase in provincial income growth. The indirect effect coefficient was significantly positive at the 1% level, which indicated that digital inclusive finance also promoted income growth in neighboring provinces, that is, it had a spatial spillover effect on income growth. The direct effect, indirect effect, and total effect coefficients for coverage breadth were all significantly positive at 1%, 5%, and 1% respectively, indicating that every 1% increase in the digital financial provincial coverage led to an income growth increase of 0.062% and a neighboring province income growth increase of 0.005%. The direct effect coefficients for usage depth and digitization were significantly positive at 1%; however, the indirect effect coefficients were all non-significantly negative, and the impact was not significant. As the digital inclusive finance and coverage breadth had positive spatial spillover effects on income growth in neighboring provinces, Hypothesis 2 was confirmed.

### 5.5. Mitigated Effects of Digital Inclusive Finance on Climate Variation

In the climate-income model, our main interest is the impact of climate variation on rural residents’ income growth and income structure. Combining the climate variation and socioeconomic data from 2011 to 2019, we performed regression and the main empirical findings are shown in Table 9. Model 38 studies the impacts of climate variation on income, whereas models 39 to 42 report results for the impacts of climate variation on income structure. The result for model 38 is statistically significant at the 10% level, and the coefficient is negative, which indicates that higher climate variation is associated with lower income. A temperature variation that is one standard deviation higher can lead to an average reduction in income of 116%. Models 39 shows that the coefficient of the climate variation variable is negative and statistically significant at the 1% level. This indicates that, the main effect comes from operating income. Harsh weather conditions increase the agricultural production costs and total inputs and reduce outputs, thus leading to lower agricultural operating income.

For the moderating effect of digital inclusive finance, we divided digital inclusive finance into three groups: low level of digital inclusive finance, medium level of digital inclusive finance and high level of digital inclusive finance, and tested the effects of climate variation in each group. The results are reported in Table 10. Model 43 shows that the coefficient of the climate variation variable is negative and is statistically significant at the 1% level, indicating that climate variation curbs the rural residents’ income growth at the low development level of digital inclusive finance. In the medium and high-level groups, the climate variation impacts positively on the income. Especially, in the high-level group, the coefficient of the climate variation variable is positive and is statistically significant at the 5% level. These results suggest that high development level of digital inclusive finance can moderate the negative impact of climate variation on rural residents’ income. Although climate variation will reduce rural residents’ income and increase their risks, digital inclusive finance mitigates this negative effect significantly, further verifying the role of digital finance in moderating climate variation.

## 6. Conclusions and Discussion

Extreme weather conditions caused by climate variation directly and negatively impact agricultural production. As the main source of agricultural production, rural residents are more affected by the reduction of income and lack of financial support due to climate variation. Using panel data from 31 provinces in China from 2011 to 2019 and a digital inclusive finance index developed by Peking University, together with historical temperature data, this study explores the role of digital inclusive finance in promoting income growth among China’s rural residents under climate variation risk. The empirical analysis of the total digital inclusive finance effect on rural resident income growth found that overall, digital inclusive finance development and all its dimensions, coverage breadth, usage depth, and digitization level, promoted rural resident income growth, with the estimation results using a two-stage least squares method (2SLS) to control endogeneity proving these findings to be robust. The income structure results showed that digital inclusive finance and all its dimensions promoted rural resident operating, wage, and transfer income growth, which clearly showed that digital inclusive finance had a comprehensive impact on rural resident income growth. The heterogeneity results found that digital inclusive finance promoted rural resident income growth in all regions but was stronger in the central and western regions.

As the digital economy develops, cyberspace is expected to become an important part of production, and as the connections between regions become closer, the geographical space and administrative barriers become smaller, and the resource flows between provinces increase. Therefore, a spatial econometric model was used to analyze the spatial spillover effects of digital inclusive finance on rural resident income growth, from which it was found that there were significant spatial agglomeration characteristics for rural resident income growth, digital inclusive finance, coverage, depth of use, and digitization, which confirmed that digital inclusive finance availability had a significant spatial spillover effect between provinces.

Finally, we further verified the role of digital inclusive finance in moderating climate variation and found that, although climate variation reduces rural residents’ income and increases their risks, digital inclusive finance will mitigate this negative effect significantly, which is beneficial to poverty alleviation and rural revitalization.

These results indicated that to adapt climate variation, alleviate poverty and promote rural revitalization, digital information infrastructure construction, financial risk prevention, digital financial knowledge availability, and e-commerce popularization would be practical paths for the optimization of inclusive finance development in rural areas. 

First, the rural information infrastructure is quite different in developing countries such as China. In some rural areas, as Internet infrastructure is poor, power supplies unstable, and logistics networks underdeveloped, the rural digital finance and e-commerce development is restricted. Therefore, it is necessary to strengthen mobile communication networks in rural areas to increase network coverage and provide a safe, convenient, cost-effective network environment.

Second, in most developing countries, most rural residents lack basic financial and Internet knowledge. Therefore, financial institutions need to promote mobile and online payments in rural areas, digital finance and agricultural supply chains need to be explored, and e-commerce companies need to be encouraged to improve rural area e-commerce systems.

Because of the imbalances in China’s rural populations, the left-behind elderly, children and other groups have poor knowledge of digital financial risk prevention and control; therefore, small financial service institutions such as local rural commercial banks, village banks, and loan companies need to strengthen their internal institutions to prevent information leakage and protect the rights and interests of their customers. Relevant departments also need to develop financial products that meet the wealth management needs of rural residents.

The existing research primarily focuses on the impact of digital inclusive finance on economic growth. The empirical results that the digital inclusive finance promoted rural residents’ income growth and higher climate change reduced the rural residents’ income growth are consistent with existing research. However, there is little research on the relationship of digital financial inclusion and farmers’ ability to minimize the risks of climate change. To fill this gap, this study applies the empirical analysis and finds that digital inclusive finance will mitigate the negative effect of climate variation on rural residents’ income growth. Nevertheless, there is some limitations in this study. Due to lack of data, the digital financial indicator used in the empirical part of this article is the “Peking University Digital inclusive finance Index of China”, lacking systematic data on rural digital finance and data after 2020. China’s digital finance is in a period of rapid development, and many problems have not been fully exposed, so the conclusions and implications are insufficient foresight. These could be well addressed in future research.

## Figures and Tables

**Figure 1 ijerph-19-08280-f001:**
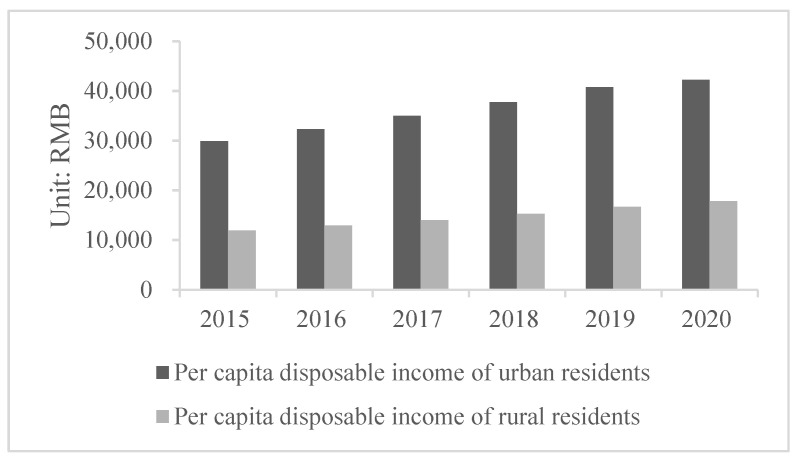
Chinese urban and rural resident per capita disposable income from 2015 to 2020 (Data source: National Bureau of Statistics of China).

**Figure 2 ijerph-19-08280-f002:**
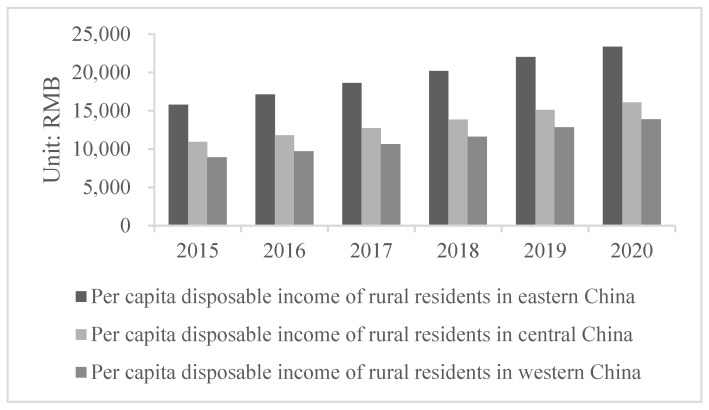
Rural resident per capita disposable income in different Chinese regions from 2015 to 2020 (Data source: National Bureau of Statistics of China).

**Figure 3 ijerph-19-08280-f003:**
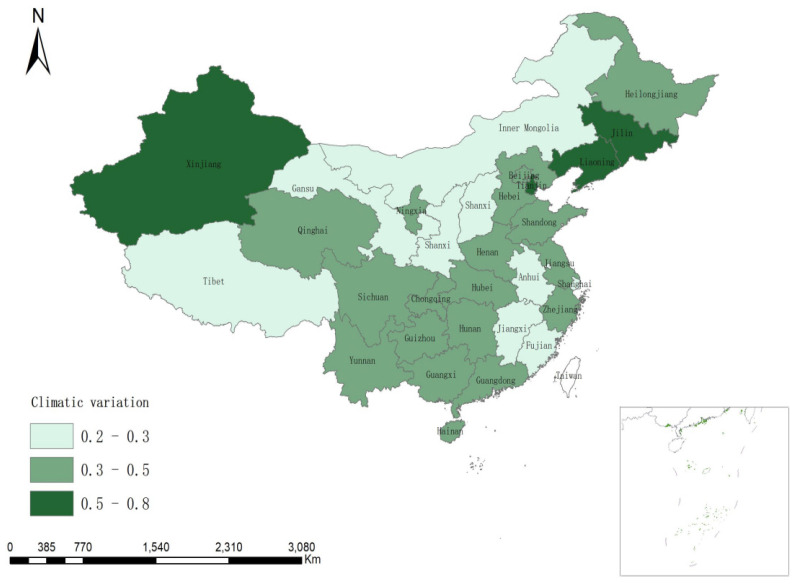
Standard deviation of the average temperature in provinces in 2019.

**Figure 4 ijerph-19-08280-f004:**
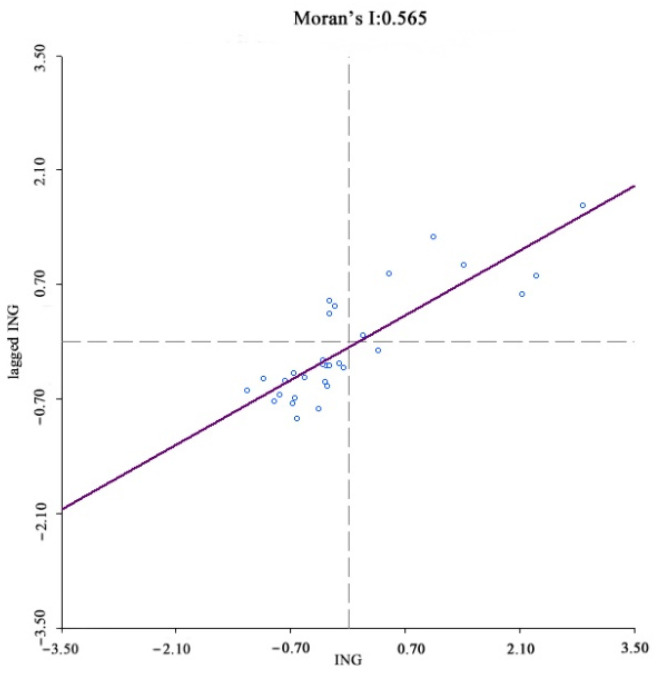
Global Moran’s I of rural residents’ income growth.

**Figure 5 ijerph-19-08280-f005:**
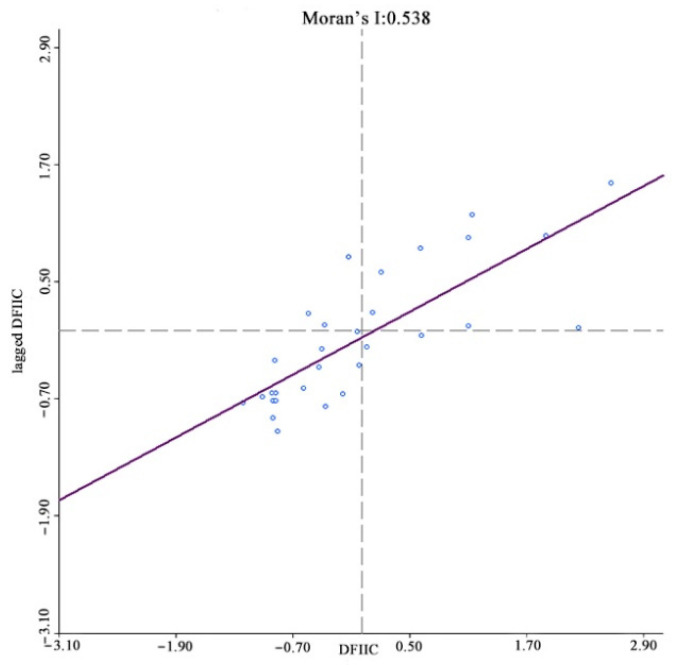
Global Moran’s I of digital inclusive finance growth.

**Figure 6 ijerph-19-08280-f006:**
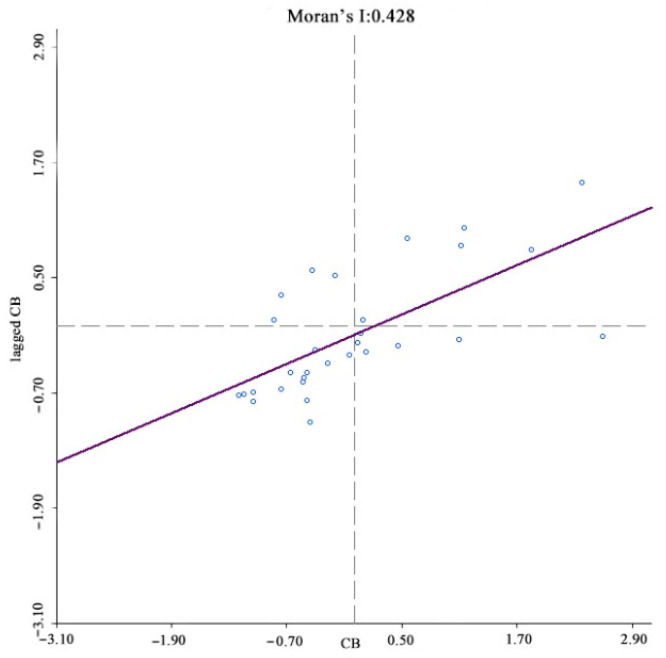
Global Moran’s I of coverage breadth growth.

**Figure 7 ijerph-19-08280-f007:**
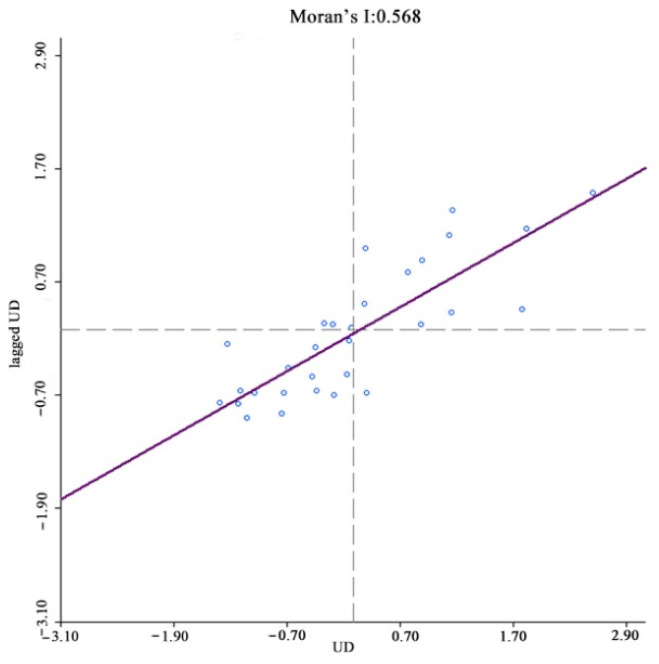
Global Moran’s I of usage depth growth.

**Figure 8 ijerph-19-08280-f008:**
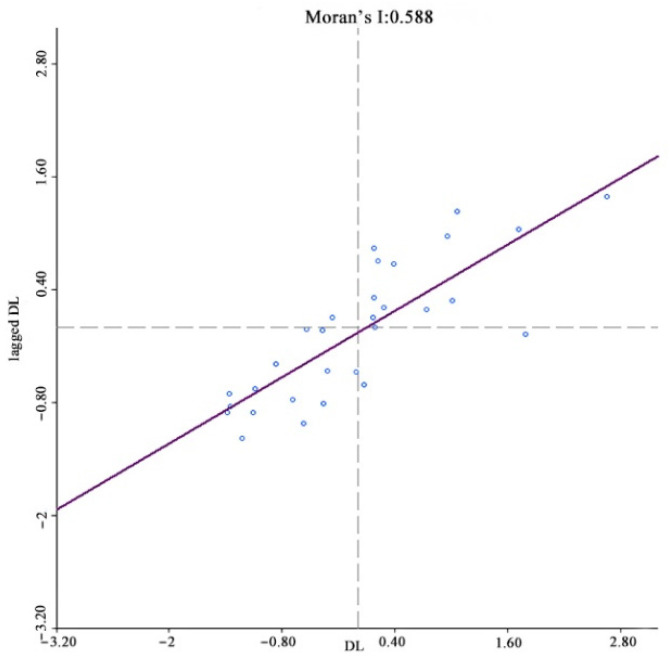
Global Moran’s I of digitization level growth.

**Table 1 ijerph-19-08280-t001:** Variable definitions.

Variables	Mark	Meaning	Details
Explained variables	lnIN	Rural residents’ income growth	Logarithm of per capita disposable income of rural residents
	lnOI	Operating income	Logarithm of per capita operating income
	lnWI	Wage income	Logarithm of per capita wage income
	lnPI	Property income	Logarithm of per capita property income
	lnTI	Transfer income	Logarithm of per capita transfer income
Core explanatory variables	lnDFIIC	Digital inclusive finance	Logarithm of the Peking University Digital inclusive finance Index of China
	lnCB	Coverage breadth	Logarithm of coverage-breadth index
	lnUD	Usage depth	Logarithm of usage-depth index
	lnDL	Digitization level	Logarithm of digitization-level index
Control variables	AMA	Agricultural machinery usage	The ratio of the total power of agricultural machinery to the planting area of crops
	FU	Fertilizer usage	The ratio of agricultural fertilizer usage to crop planting area
	GSA	Government support for agriculture	The ratio of fiscal expenditure on agriculture, forestry and water affairs to agricultural GDP
	FSA	Financial support for agriculture	The ratio of agricultural loans to agricultural GDP
	TRADE	Trade openness	The ratio of total import and export to GDP
	lnperGDP	GDP per capita	Logarithm of GDP per capita
	UR	Urbanization usage	Urbanization rate
	TI	The proportion of tertiary industry	The proportion of tertiary industry

**Table 2 ijerph-19-08280-t002:** Descriptive Statistics.

Variables	Obs	Mean	Std.Dev.	Min	Max
lnIN	279	9.28	0.408	8.214	10.385
lnOI	279	8.287	0.405	6.726	8.99
lnWI	279	8.303	0.654	6.633	9.947
lnPI	279	5.511	0.741	3.661	7.64
lnTI	279	7.403	0.656	5.834	9.137
lnDFIIC	279	5.143	0.679	2.786	6.017
lnCB	279	4.979	0.852	0.673	5.952
lnUD	279	5.126	0.649	1.911	6.087
lnDL	279	5.458	0.717	2.026	6.136
AMA	279	0.683	0.352	0.264	2.463
FU	279	0.036	0.013	0.011	0.075
GSA	279	0.562	0.81	0.126	5.142
FSA	279	0.625	1.247	0.046	11.933
TRADE	279	0.041	0.047	0.002	0.24
lnperGDP	279	10.773	0.44	9.656	11.986
UR	279	0.574	0.132	0.228	0.896
TI	279	0.469	0.096	0.297	0.835

**Table 3 ijerph-19-08280-t003:** Impact of inclusive digital finance on rural resident income growth.

	(1)	(2)	(3)	(4)	(5)	(6)	(7)	(8)
	Current Explanatory Variable				Lagging Explanatory Variable			
lnDFIIC	0.137 ***				0.097 ***			
	(0.008)				(0.007)			
lnCB		0.088 ***				0.061 ***		
		(0.007)				(0.005)		
lnUD			0.111 ***				0.054 ***	
			(0.009)				(0.006)	
lnDL				0.087 ***				0.079 ***
				(0.006)				(0.007)
AMA	0.003	−0.016	0.036	0.002	−0.019	−0.032	−0.018	0.010
	(0.029)	(0.033)	(0.034)	(0.032)	(0.024)	(0.026)	(0.028)	(0.027)
FU	−1.286	−0.541	−1.643	0.304	−0.908	−0.440	0.252	−1.456
	(0.919)	(1.054)	(1.110)	(1.034)	(0.791)	(0.878)	(0.926)	(0.897)
GSA	0.054 ***	0.050 ***	0.049 ***	0.067 ***	0.052 ***	0.049 ***	0.061 ***	0.049 ***
	(0.010)	(0.012)	(0.012)	(0.011)	(0.010)	(0.011)	(0.011)	(0.011)
FSA	0.005 **	0.005 **	0.009 ***	0.005 **	0.002	0.002	0.003	0.005 ***
	(0.002)	(0.002)	(0.002)	(0.002)	(0.002)	(0.002)	(0.002)	(0.002)
TRADE	−1.067 ***	−1.201 ***	−0.962 ***	−0.360	−0.844 ***	−0.852 ***	−0.244	−0.704 ***
	(0.279)	(0.323)	(0.335)	(0.313)	(0.237)	(0.265)	(0.271)	(0.266)
lnperGDP	0.175 ***	0.235 ***	0.263 ***	0.184 ***	0.294 ***	0.368 ***	0.347 ***	0.387 ***
	(0.036)	(0.041)	(0.042)	(0.042)	(0.038)	(0.041)	(0.045)	(0.041)
UR	2.874 ***	2.755 ***	2.917 ***	3.751 ***	2.194 ***	1.993 ***	2.696 ***	2.123 ***
	(0.219)	(0.254)	(0.264)	(0.247)	(0.206)	(0.229)	(0.247)	(0.232)
TI	0.763 ***	1.003 ***	0.854 ***	0.790 ***	0.682 ***	0.864 ***	0.734 ***	0.746 ***
	(0.090)	(0.101)	(0.108)	(0.103)	(0.075)	(0.081)	(0.089)	(0.084)
Constant	4.738 ***	4.301 ***	3.838 ***	4.259 ***	4.184 ***	3.603 ***	3.427 ***	3.280 ***
	(0.306)	(0.348)	(0.353)	(0.340)	(0.325)	(0.350)	(0.369)	(0.342)
Provincial fixed effect	YES	YES	YES	YES	YES	YES	YES	YES
Observations	279	279	279	279	248	248	248	248
Within R-squared	0.982	0.976	0.974	0.977	0.985	0.981	0.979	0.981
Hausman test	0.000	0.000	0.000	0.000	0.000	0.000	0.000	0.000

Notes: Standard errors in parentheses. *** *p* < 0.01, ** *p* < 0.05.

**Table 4 ijerph-19-08280-t004:** Digital inclusive finance and rural resident income structure.

	(9)	(10)	(11)	(12)
	Operating Income	Wage Income	Property Income	Transfer Income
lnDFIIC	0.094 *** (0.011)	0.104 *** (0.020)	−0.136 *** (0.045)	0.497 *** (0.050)
Control variables	YES	YES	YES	YES
Provincial fixed effect	YES	YES	YES	YES
Observations	279	279	279	279
Within R-squared	0.929	0.891	0.598	0.851
Hausman test	0.000	0.000	0.003	0.001

Notes: Standard errors in parentheses. *** *p* < 0.01.

**Table 5 ijerph-19-08280-t005:** Coverage breadth, usage depth, digitization level, and rural resident income structure.

**Explained Variables: Operating Income**				**Explained Variables: Wage Income**		
	**(13)**	**(14)**	**(15)**	**(16)**	**(17)**	**(18)**
lnCB	0.060 *** (0.010)			0.065 *** (0.015)		
lnUD		0.086 *** (0.012)			0.077 *** (0.021)	
lnDL			0.061 *** (0.008)			0.064 *** (0.015)
**Explained Variables: Property Income**				**Explained Variables: Transfer Income**		
	**(19)**	**(20)**	**(21)**	**(22)**	**(23)**	**(24)**
lnCB	−0.055 (0.034)			0.312 *** (0.039)		
lnUD		−0.120 *** (0.045)			0.348 *** (0.055)	
lnDL			−0.120 *** (0.032)			0.321 *** (0.037)

Notes: Standard errors in parentheses. *** *p* < 0.01; all the results above have controlled provincial and control variables.

**Table 6 ijerph-19-08280-t006:** Robustness check (IV).

	2SLS			
	(25)	(26)	(27)	(28)
lnDFIIC	0.288 *** (0.017)			
lnCB		0.220 *** (0.016)		
lnUD			0.406 *** (0.051)	
lnDL				0.166 *** (0.015)
Control variables	YES	YES	YES	YES
Provincial fixed effect	YES	YES	YES	YES
Observations	248	248	248	248
IV (Lagging)	YES	YES	YES	YES
Within R-squared	0.984	0.982	0.943	0.973
Hausman test	0.000	0.000	0.000	0.823

Notes: Standard errors in parentheses. *** *p* < 0.01.

**Table 7 ijerph-19-08280-t007:** Regional heterogeneous results.

	Current Explanatory Variables			Lagging Explanatory Variable		
	Eastern Area (29)	Central Area (30)	Western Area (31)	Eastern Area (32)	Central Area (33)	Western Area (34)
lnDFIIC	0.111 *** (0.015)	0.142 *** (0.013)	0.135 *** (0.013)	0.062 *** (0.013)	0.116 *** (0.010)	0.109 *** (0.010)
Control variables	YES	YES	YES	YES	YES	YES
Provincial fixed effect	YES	YES	YES	YES	YES	YES
Observations	98	72	108	88	64	96
Hausman test	0.000	0.000	0.000	0.000	0.000	0.000
Within R-squared	0.983	0.990	0.987	0.986	0.994	0.991

Notes: Standard errors in parentheses. *** *p* < 0.01.

**Table 8 ijerph-19-08280-t008:** SDM results.

	Spatial Weight Matrix		
	Direct Effect (35)	Indirect Effect (36)	Total Effect (37)
lnDFIIC	0.172 *** (0.019)	0.073 *** (0.025)	0.245 *** (0.014)
lnCB	0.062 *** (0.017)	0.005 ** (0.021)	0.067 *** (0.012)
lnUD	0.100 *** (0.022)	−0.024 (0.024)	0.075 *** (0.014)
lnDL	0.074 *** (0.015)	−0.001 (0.015)	0.073 *** (0.011)

Notes: Standard errors in parentheses. *** *p* < 0.01, ** *p* < 0.05.

**Table 9 ijerph-19-08280-t009:** Climate variation and rural residents’ income.

	Total Income	Operating Income	Wage Income	Property Income	Transfer Income
	(38)	(39)	(40)	(41)	(42)
climate	−0.116 *	−0.148 ***	−0.084	−0.169	−0.165
	(−1.91)	(−3.05)	(−0.96)	(−1.19)	(−0.85)
Control variables	YES	YES	YES	YES	YES
Provincial fixed effect	YES	YES	YES	YES	YES
Observations	279	279	279	279	279
Within R-squared	0.869	0.845	0.757	0.512	0.726

Notes: Standard errors in parentheses. *** *p* < 0.01, * *p* < 0.1.

**Table 10 ijerph-19-08280-t010:** Moderating effect of digital inclusive finance.

	Low Level Group	Medium Level Group	High Level Group
	(43)	(44)	(45)
climate	−0.149 *	0.066	0.441 **
	(0.083)	(0.091)	(0.203)
Control variables	YES	YES	YES
Provincial fixed effect	YES	YES	YES
Observations	93	93	93
Within R-squared	0.949	0.840	0.738

Notes: Standard errors in parentheses. ** *p* < 0.05, * *p* < 0.1.

## Data Availability

Data used in this study are available upon request.

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
