# Peer review of "Does Digital Inclusive Finance Mitigate the Negative Effect of Climate Variation on Rural Residents’ Income Growth in China?"

_ijerph, 2022, doi:10.3390/ijerph19148280_

Round 1

Reviewer 1 Report

The interesting analysis is presented in the paper. The research gap relates to the lack of understanding of the impact of the digital finance as a tool of inclusive finance approach on the rural development and welfare (income) of farmers in the rural areas, which would help to overcome the negative consequences of climate variations.

The paper presents correct research methodology, clear results, and the conclusions made on the basis of the outcome of the research.

Nevertheless, several remarks are worth to be mentioned:

- The data of the period 2011-2019 help to conclude in favor of the authors' idea, but the lack of study of 2020-2022 permits to suspect that in the next 3 years the tendency could differ from the years after the global financial collapse of 2008-2009. It would be reasonable to mention, why only the period of 2011-2019 is taked into account.

- the impact of the digital technologies' growth on the inclusive finance and access to information and to credits is clear, but the important problem all over the world is the availability  in the rural areas of telecommunication' infrastructure and access to internet. Without this access, the digital finance can not help the inclusion of rural population. The argumentation in lines 100-104 should be developed and completed with, at least, a note about the accessibility. The paragraph on the lines 207-215 could tell us about this limitation, but the text only gives some declarations about "no time and space limitations", thus, there are some limitations worthy to be mentioned.

For example: "digital finance provides a 209 convenient, safe, instant, digital payment network." (lines 209-210) - this sentence is not completely correct, because the infrastructure provides the safe... network, the personal contacts provide the human and social networking, and the finance itself does not create any network.

- a similar note concerns the lines 166-168, the capital and finance are significant, but the implementation of innovative methods of management and of new agrictultural technologies (digital twins, drones, camerar, etc.) seem to play crucial dramatic direct role. The text should mention that the finance plays the role of a tool to help increasing level of education, of high technologies' use, of management. The reader has to guess again this complex link between the welfare of rural inhabitants and the access to credits (because this correlation can be negative - too much credits can ruine the welfare in rural areas), the authors should prove their idea themselves and not to rely on the smart thinking of the readers (who can make opposite conclusions).

- there is a lack of logic - lines 105-106: "Digital financial inclusion continues to develop rapidly in China. However, little attention has been paid to promoting rural residents’ income growth under climate variation." - Both sentences are correct, but there is no link between them in the paragraph.

- section 2.2. treats with the very general conclusions that concern all agents (not only rural population) and all finance (not only inclusive finance). If an agent has access to a resource, the possessed resource helps this agent to get larger and faster access to other resources, it is called poverty trap (line 55 of the article, reference [9]). This conclusion is very general and requires the more clear explanation related to the subject of the article - the inclusive finance covers the poverty trap, it is the essence of the inclusive finance' definition, the authors do not need to prove it (Nobel prize was awarded for this concet development).

- many repeating sentences can be detected. The essential ideas could be formulated in different reasoning and form (climate - lines 35-36; 145;  517; poverty and digital finance - lines 59-60; 62; 80-81; 84-85; 101-103; 149; 152; 154; 175-176).

- it seems a typo in the sentence (lines 205-206): "Timely financial information can be easily found on the, which reduces the costs of learning about financial services." (a word is missed before the comma?)

Despite these remarks, the research is correct, the significance of the conclusions is high, and the article can be published because it represents interesting outcomes to enhance the sustainable development of the whole area and the rural-urban areas' populations balance.

Author Response

Dear anonymous reviewers,

Manuscript ID: ijerph-1776119

Manuscript Title: Does digital inclusive finance mitigate the negative effect of climate variation on rural residents’ income growth in China?

Firstly, we appreciate the suggestions of the anonymous reviewers. Based on these suggestions, the paper has been revised carefully. Our specific responses to the editors and reviewers’ comments are as follows.

 Thank you again for your valuable comments and suggestions, they are our valuable experience, which have helped us a lot to improve the quality of our study, we have learned a lot!

Responses to Reviewer 1:

Comment: The interesting analysis is presented in the paper. The research gap relates to the lack of understanding of the impact of the digital finance as a tool of inclusive finance approach on the rural development and welfare (income) of farmers in the rural areas, which would help to overcome the negative consequences of climate variations. The paper presents correct research methodology, clear results, and the conclusions made on the basis of the outcome of the research. Nevertheless, several remarks are worth to be mentioned.

Reply: Thank you for your helpful comments and suggestions. Under your guidance, we revised the manuscript and learned a lot. In the revised manuscript, we have made modifications one by one according to the comments of the reviewer, and all the modifications are marked as yellow in the revised manuscript.

(1) Comment: The data of the period 2011-2019 help to conclude in favor of the authors' idea, but the lack of study of 2020-2022 permits to suspect that in the next 3 years the tendency could differ from the years after the global financial collapse of 2008-2009. It would be reasonable to mention, why only the period of 2011-2019 is taken into account.

(1) Reply: Thank you for your helpful suggestions. As the reviewer said, we haven't explained why the period of 2011-2019 was chosen. Before 2011, digital inclusive finance in most parts of China did not occur, so the data of Peking University Digital Inclusive Finance Index of China (PKU-DFIIC) is available as early as 2011. The Peking University Digital Inclusive Finance Index is updated to 2020. Furthermore, all other data were taken from the statistical yearbooks, which are officially released by the Bureau of Statistics. The latest data is for 2020. So far, we can get data from 2011 to 2020. Then to ensure the robustness of the results, that is, whether there will be different trends in the impact of digital inclusive finance on rural residents' income growth after 2019, we also used the panel data from 2011 to 2020 to test (Appendix A) (L608-609), and found that the results are consistent with the period of 2011-2019. Therefore, from the perspectives of digital finance development level, data availability and results robustness, we add one paragraph in the revised manuscript to explain the reasons for choosing the data from 2011 to 2019 (L397-405). All the modifications are marked as yellow in the revised manuscript.

(2) Comment: The impact of the digital technologies' growth on the inclusive finance and access to information and to credits is clear, but the important problem all over the world is the availability in the rural areas of telecommunication' infrastructure and access to internet. Without this access, the digital finance can not help the inclusion of rural population. The argumentation in lines 100-104 should be developed and completed with, at least, a note about the accessibility.

(2) Reply: Thank you for your helpful suggestions. As the reviewer said, without the access of telecommunication' infrastructure and internet, the digital finance cannot help the rural residents. The argumentation about accessibility should be developed and completed. In the section “1. Introduction” of the revised manuscript, we add the argumentation about accessibility (L98-111). The specific content is as follows: “Digital inclusive finance development relies on digital infrastructure and consumer awareness……whether digital inclusive finance mitigate the negative effect of climate variation on rural residents’ welfare has a certain reference value for countries in which there are large rural-urban income gaps.” All the modifications are marked as yellow in the revised manuscript.

(3) Comment: The paragraph on the lines 207-215 could tell us about this limitation, but the text only gives some declarations about "no time and space limitations", thus, there are some limitations worthy to be mentioned. For example: "digital finance provides a 209 convenient, safe, instant, digital payment network." (lines 209-210) - this sentence is not completely correct, because the infrastructure provides the safe... network, the personal contacts provide the human and social networking, and the finance itself does not create any network.

(3) Reply: Thank you for your helpful suggestions. As the reviewer said, the digital finance doesn’t play crucial direct role in providing a convenient, safe, instant, digital payment network. So, we delete the sentence “Therefore, digital finance provides a convenient, safe, instant, digital payment network”. Moreover, according to the comments of the reviewer, we add the discussion about limitations (205-215) in the third paragraph of section “3. Theoretical hypothesis”. The specific content is as follows: “Compared with traditional payments, online payments have no time and space limitations.……If their synergy fails, the digital finance may be ineffective, and even emerge new financial exclusion.” All the modifications are marked as yellow in the revised manuscript.

(4) Comment: A similar note concerns the lines 166-168, the capital and finance are significant, but the implementation of innovative methods of management and of new agricultural technologies (digital twins, drones, camerar, etc.) seem to play crucial dramatic direct role. The text should mention that the finance plays the role of a tool to help increasing level of education, of high technologies' use, of management. The reader has to guess again this complex link between the welfare of rural inhabitants and the access to credits (because this correlation can be negative - too much credits can ruine the welfare in rural areas), the authors should prove their idea themselves and not to rely on the smart thinking of the readers (who can make opposite conclusions).

(4) Reply: Thank you for your helpful suggestions. As the reviewer said, we didn’t explain the link between the digital inclusive finance and the welfare of rural residents clearly. Digital inclusive finance improves the circulation of the rural financial system and the new technologies adoption of rural residents, and promotes innovation of management and new technologies and digital upgrade of rural industries, which indirectly increases rural residents’ income and welfare. In the revised manuscript, and in the first paragraph of the section “2.2 Digital inclusive finance and rural resident income growth” (L156-166), we rewrite it to explain the complex link. The specific content is as follows: “Digital inclusive finance, which is the combination of digital technology and financial services.……improve their risk resilience, and indirectly increase their income levels.” All the modifications are marked as yellow in the revised manuscript.

(5) Comment: There is a lack of logic - lines 105-106: "Digital financial inclusion continues to develop rapidly in China. However, little attention has been paid to promoting rural residents’ income growth under climate variation." Both sentences are correct, but there is no link between them in the paragraph.

(5) Reply: Thank you for your helpful suggestions. As the reviewer said, these two sentences are correct, but there is no logic link between them. In the revised manuscript, and in the last paragraph of the section “1. Introduction”, we change the sentences “Digital financial inclusion continues to develop rapidly in China. However, little attention has been paid to promoting rural residents’ income growth under climate variation.” to “Digital inclusive finance continues to develop rapidly in China and its impact on rural residents’ welfare has been widely concerned. Little attention has been paid to promoting rural residents’ income growth under climate variation.” (L112-114). All the modifications are marked as yellow in the revised manuscript.

(6) Comment: Section 2.2. treats with the very general conclusions that concern all agents (not only rural population) and all finance (not only inclusive finance). If an agent has access to a resource, the possessed resource helps this agent to get larger and faster access to other resources, it is called poverty trap (line 55 of the article, reference [9]). This conclusion is very general and requires the more clear explanation related to the subject of the article - the inclusive finance covers the poverty trap, it is the essence of the inclusive finance' definition, the authors do not need to prove it (Nobel prize was awarded for this concept development).

(6) Reply: Thank you for your helpful suggestions. As the reviewer said, the conclusions of section “2.2 Financial development and rural resident income growth” is very general and not relevant to the subject. In the revised manuscript, we drop this section and add more researches about the link between digital inclusive finance and rural resident income growth (L156-166) in the new section “2.2 Digital inclusive finance and rural resident income growth”. All the modifications are marked as yellow in the revised manuscript.

(7) Comment: Many repeating sentences can be detected. The essential ideas could be formulated in different reasoning and form (climate - lines 35-36; 145; 517; poverty and digital finance - lines 59-60; 62; 80-81; 84-85; 101-103; 149; 152; 154; 175-176).

(7) Reply: Thank you for your helpful suggestions. According to the comments of the reviewer, for climate expression, we delete the repeating sentences “Overall, climate variation continues to reduce the rural residents’ income” (L145 of the original manuscript) and “The ecological environment impacts the socioeconomic development deeply.” (L517 of the original manuscript) in the revised manuscript. For poverty and digital finance, we delete the repeating sentences of L149, L152, L154 in the original manuscript. Then we rewrite other sentences (L56-58), (L81-84) in the section “1. Introduction” and (L172) in the section “2. Literature review” in the revised manuscript. The modifications are marked as yellow in the revised manuscript.

(8) Comment: It seems a typo in the sentence (lines 205-206): "Timely financial information can be easily found on the, which reduces the costs of learning about financial services." (a word is missed before the comma?)

(8) Reply: Thank you for your helpful suggestions. As the reviewer said, we missed a word before the comma in this sentence. In the revised manuscript, we add the word “internet” to this sentence (L201-202). The specific content is as follows: “Timely financial information can be easily found on the internet, which reduces the costs of learning about financial services.” All the modifications are marked as yellow in the revised manuscript.

(9) Comment: Despite these remarks, the research is correct, the significance of the conclusions is high, and the article can be published because it represents interesting outcomes to enhance the sustainable development of the whole area and the rural-urban areas' populations balance.

(9) Reply: Special thanks to you for your valuable comments and suggestions, which have helped us further improve the quality of our study! We learned a lot! Thanks again!

Reviewer 2 Report

First of all, thank you for submitting the article for review. I have some comments:

1. In formula 2, not all variables are described / explained

2. Fig 4 to 8 the axis descriptions are illegible

3. The chapter with the discussion of the results is missing, in particular the comparison of the results to other national / international studies, please complete this part.

Author Response

Dear anonymous reviewers,

Manuscript ID: ijerph-1776119

Manuscript Title: Does digital inclusive finance mitigate the negative effect of climate variation on rural residents’ income growth in China?

Firstly, we appreciate the suggestions of the editors and anonymous reviewers. Based on these suggestions, the paper has been revised carefully. Our specific responses to the editors and reviewers’ comments are as follows.

Thank you for your valuable comments and suggestions, they are our valuable experience, which have helped us a lot to improve the quality of our study, we have learned a lot!

Responses to Reviewer 2:

Comment: First of all, thank you for submitting the article for review. I have some comments.

Reply: Thank you for your helpful comments and suggestions. Under your guidance, we revised the manuscript and learned a lot. In the revised manuscript, we have made modifications one by one according to the comments of the reviewer, and all the modifications are marked as yellow in the revised manuscript.

(1) Comment: In formula 2, not all variables are described / explained.

(1) Reply: Thank you for your helpful suggestions. As the reviewer said, we didn’t describe all the variables. In the revised manuscript, we add one paragraph (L330-335) after formula (2) to clearly describe the variables not explained before, such as variable . The specific content are as follows: “Where  is spatial adjacency matrix,  is a constant term,  is the fixed effect, and  is an error term……The definitions of other variables are the same as formula (1)”. All the modifications are marked as yellow in the revised manuscript.

(2) Comment: Fig 4 to 8 the axis descriptions are illegible.

(2) Reply: Thank you for your helpful suggestions. As the reviewer said, Figure 4 to 8 the axis descriptions are illegible. We redraw the Figure 4 to 8 and make the axis descriptions clear (L473-485). All the modifications are marked as yellow in the revised manuscript.

(3) Comment: The chapter with the discussion of the results is missing, in particular the comparison of the results to other national / international studies, please complete this part.

(3) Reply: Thank you for your helpful suggestions. According to the comments of the reviewer, in the section “6. Conclusions and discussions” of the revised manuscript, we add one discussion paragraph (L595-607) to compare results of this study and existing researches, and propose some limitations in this study. The specific content are as follows: “The existing research primarily focuses on the impact of digital inclusive finance on economic growth.……These could be well addressed in future research”. All the modifications are marked as yellow in the revised manuscript.

Special thanks to you for your valuable comments and suggestions, which have helped us further improve the quality of our study! We learned a lot! Thanks again!

Round 2

Reviewer 2 Report

Thank you for yours response. I accept the article for publication, I have no further comments for my part.

This manuscript is a resubmission of an earlier submission. The following is a list of the peer review reports and author responses from that submission.